# The Effect of Reduced Fibrinogen on Cerebrovascular Permeability during Traumatic Brain Injury in Fibrinogen Gene Heterozygous Knockout Mice

**DOI:** 10.3390/biom14040385

**Published:** 2024-03-22

**Authors:** Nurul Sulimai, Jason Brown, David Lominadze

**Affiliations:** 1Department of Surgery, College of Medicine, University of South Florida Morsani, Tampa, FL 33612, USA; nurulsulimai@usf.edu (N.S.); jasonb3@usf.edu (J.B.); 2Department of Molecular Pharmacology and Physiology, College of Medicine, University of South Florida Morsani, Tampa, FL 33612, USA

**Keywords:** fibrinogen deposition, neuroinflammation and nuclear factor-кB

## Abstract

Vascular contribution to cognitive impairment and dementia (VCID) is a term referring to all types of cerebrovascular and cardiovascular disease-related cognitive decline, spanning many neuroinflammatory diseases including traumatic brain injury (TBI). This becomes particularly important during mild-to-moderate TBI (m-mTBI), which is characterized by short-term memory (STM) decline. Enhanced cerebrovascular permeability for proteins is typically observed during m-mTBI. We have previously shown that an increase in the blood content of fibrinogen (Fg) during m-mTBI results in enhanced cerebrovascular permeability. Primarily extravasated via a transcellular pathway, Fg can deposit into the parenchyma and exacerbate inflammatory reactions that can lead to neurodegeneration, resulting in cognitive impairment. In the current study, we investigated the effect of a chronic reduction in Fg concentration in blood on cerebrovascular permeability and the interactions of extravasated Fg with astrocytes and neurons. Cortical contusion injury (CCI) was used to generate m-mTBI in transgenic mice with a deleted Fg γ chain (Fg γ+/−), resulting in a low blood content of Fg, and in control C57BL/6J wild-type (WT) mice. Cerebrovascular permeability was tested in vivo. Interactions of Fg with astrocytes and neurons and the expression of neuronal nuclear factor-кB (NF-кB) were assessed via immunohistochemistry. The results showed that 14 days after CCI, there was less cerebrovascular permeability, lower extravascular deposition of Fg, less activation of astrocytes, less colocalization of Fg with neurons, and lower expression of neuronal pro-inflammatory NF-кB in Fg γ+/− mice compared to that found in WT mice. Combined, our data provide strong evidence that increased Fg extravasation, and its resultant extravascular deposition, triggers astrocyte activation and leads to potential interactions of Fg with neurons, resulting in the overexpression of neuronal NF-кB. These effects suggest that reduced blood levels of Fg can be beneficial in mitigating the STM reduction seen in m-mTBI.

## 1. Introduction

Traumatic brain injury (TBI)-induced inflammation causes alterations in the blood–brain barrier (BBB), resulting in increased permeability for blood components. This increased BBB permeability could persist for years after the head trauma [1]. TBI is an inflammatory disease that is associated with an increased blood level of fibrinogen (Fg), called hyperfibrinogenemia (HFg) [2]. Elevated blood levels of Fg were associated with an increased risk of cerebrovascular lesions [3]. There have been incidents of extravascular deposition of fibrin in the brains of human patients with TBI who survived 18 years after a fall trauma [1,4]. Fibrin deposits were observed in 35 out of 151 human brain tissue specimens in and around contusion lesions [4]. Fg deposits in the brain have been associated with inflammation and neuronal loss in acute human TBI [4].

The BBB protects neurons from factors with adverse effects originating in the systemic circulation. During TBI, BBB impairment facilitates the entry of blood-derived products, such as Fg, into the brain tissue. Plasma Fg is primarily synthesized in the liver, with the assembly of its three chains (Aα, Bβ and γ) into a six-chain dimer occurring in the lumen of the endoplasmic reticulum of hepatocytes [5,6,7]. Although it has been shown that neurons generate all three chains of Fg, the synthesis of the whole Fg protein has not been documented [8]. Therefore, Fg and/or fibrin deposits in extravascular brain tissue may only happen in the event of severe or acute TBI [4] due to the disruption of blood vessels and resultant bleeding or as a result of increased cerebrovascular permeability that results in slow extravasation of Fg [9,10]. Only after deposition and immobilization in the brain extravascular space is Fg converted to fibrin. The deposition of fibrin in the brain tissue as a result of bleeding predominantly occurs during moderate-to-severe TBI [1], while mild-to-moderate TBI (m-mTBI) is mainly characterized by increased cerebrovascular permeability for proteins in the absence of bleeding [11,12]. In fact, in mice with m-mTBI, ruptured vessels are not found 9–11 days after head injury [11]. In this model of TBI, inflammation causes HFg [2,13], and as a consequence, an increase in cerebrovascular permeability results in the extravasation of Fg, mainly via a transcellular pathway [9,14]. We found that extravasated Fg deposited in the vasculo-astrocyte interface [10] and activated astrocytes [10,15] and then neurons [15,16] through specific interactions with the astrocytic [14] and neuronal [15] receptors. We have also shown that Fg induced activation of pro-inflammatory transcription factor nuclear factor-кB (NF-кB) in cultured primary neurons [17] and increased the generation of nitric oxide (NO) and reactive oxygen species (ROS), resulting in the death of the astrocytes and neurons [16,17]. These effects of Fg were associated with a reduction in short-term memory (STM) [18]. All of these results suggest that during an inflammatory disease, such as TBI, HFg can be one of the main culprits resulting in various destructive responses leading to a reduction in memory. As Fg is mainly synthesized in the liver and circulates in the blood, its neurodegenerative effects positively fall in the category of vascular cognitive impairment (VCI). Previously, we have shown that an induced acute reduction in the Fg concentration in the blood can result in the amelioration of some inflammatory effects associated with TBI-induced HFg [18]. In this study, we explored the effect of a genetic reduction in the level of Fg on cerebrovascular permeability, on the deposition of Fg in the extravascular space, on Fg’s interaction with astrocytes and neurons, and on the activation of NF-кB, expecting amelioration in STM changes to be seen in a mouse model of m-mTBI [10,18]. To ensure a minimal increase in the blood content of Fg during TBI, we used heterozygous Fg gamma-chain knockout (Fg γ+/−) mice (a gift from Dr. Francis J. Castellino, Univ. of Notre Dame, Indiana) that carry an allele with a targeted deletion of the Fg gamma chain gene, resulting in a lower blood level of Fg [19,20].

## 2. Results

### 2.1. Blood Level of Fg in the γ+/− Mice

The genotyping of the mice showed the presence of the neomycin phosphotransferase (NEO) gene that replaced the Fg γ-chain, indicating the Fg γ+/− mice versus normal control WT mice (Figure 1). These results coincide with the data of Dr. Castellino, who developed this mouse model [19]. Although the blood level of Fg was increased in both WT and Fg γ+/− mice during TBI in comparison to that in sham-operated mice, it was lower in the Fg γ+/− mice than that in the WT mice with TBI (Figure 1a,b) (See also Appendix A) and with a sham operation (Figure 1c,d) (See also Appendix A).

The disrupted allele was amplified with forward primer 2 in the 5′-noncoding flanking region of Fg γ-chain and with reverse primer 3, located in the NEO gene, which resulted in a 0.55 kb amplicon. In wild-type (WT) mice, amplification of the Fg γ-chain allele was performed with forward primer 2 in the Fg γ-chain 5′-flank and the reverse primer 4 within the Fg γ-chain gene, resulting in a 0.42 kb amplicon.

### 2.2. Cerebrovascular Permeability in Fg γ+/− Mice-

Both Fg γ+/− and WT mice with CCI showed a higher pial venular permeability compared to that in the respective sham-operated mice (Figure 2). Chronic reduction in Fg levels in the Fg γ+/− mice significantly reduced the pial venular permeability to bovine serum albumin (BSA) labeled with fluorescein isothiocyanate (FITC-BSA) 14 days after CCI (Figure 2).

### 2.3. Extravasation of BSA and Activation of Astrocytes

CCI induced an increased extravasation of BSA in the mouse brains 14 days after CCI (Figure 3). The amount of colocalization of extravasated BSA with astrocytes was significantly lower in Fg γ+/− mice than that in WT mice after CCI (Figure 3). These results were associated with a lower increase in the glial fibrillary acidic protein (GFAP) expression in Fg γ+/− mice than that in WT mice after CCI (Figure 3) (See also Appendix A).

### 2.4. Association of Extravascularly Deposited Fg with Neurons after CCI

Immunohistochemical analyses of mouse brain samples showed that extravasated Fg was colocalized with neurons, and this Fg–neuronal association was lower in Fg γ+/− mice 14 days after CCI than that in WT mice (Figure 4) (See also Appendix A).

### 2.5. Expression of Nuclear Factor-кBp65 (NF-кBp65) and Its Colocalization with Neurons in the Brain after CCI

A higher colocalization of NF-кBp65 and neurons was found in the brains of the WT mice compared to that in the Fg γ+/− 14 days after CCI (Figure 5). Predictably, in the sham-operated mice, there was a lower colocalization of NF-кBp65 and neurons in the Fg γ+/− mice compared to that in the WT mice (Figure 5).

## 3. Discussion

Although it is well known that TBI affects memory, impairment in the STM mainly occurs during m-mTBI [21]. Contrary to moderate-to-severe or severe TBIs, which are characterized by ruptured vessels, bleeding, and the resultant accumulation of Fg in the brain [4], during m-mTBI there are fewer (or no) ruptured vessels and little to no resultant bleeding [11]. However, m-mTBI is accompanied by an enhanced deposition of Fg [4,10,14,22] and is associated with amyloid-beta (Aβ) and/or with cellular prion protein (PrP^C^) [14]. The latter has a stronger effect on memory [23,24]. Severe TBI, accompanied by vascular bleeding, results in a decrease in the blood level of Fg [25,26]. Contrary to this, m-mTBI results in an inflammation-induced HFg [2,13]. Data from this and our previous study confirm an increase in the blood level of Fg during m-mTBI [22]. The presented data predictably indicate that Fg γ+/− mice have a lower blood content of Fg. Most importantly, the blood level of Fg in these mice with TBI was still lower than that in WT mice with TBI. The enhanced extravascular deposition of Fg during m-mTBI seen in the brain of WT mice could be a result of an increased blood level of Fg itself, which is an inflammatory agent [14,27,28]. Our previous studies suggest that HFg in general, [14] and during m-mTBI [22], resulted in increased cerebrovascular permeability to Fg and its extravascular deposition [10,14,22]. The present data indicate that during m-mTBI, cerebrovascular permeability is lower in Fg γ+/− mice compared to that in WT mice. There was a greater colocalization of BSA with the astrocytes in the WT mice during TBI, indicating increased cerebrovascular permeability. In Fg γ+/− mice, cerebrovascular permeability during TBI was lower than that in WT mice with TBI. These results suggest a strong association of Fg blood levels with BBB property changes, indicating its role in cerebrovascular permeability during inflammatory diseases such as TBI.

Under normal circumstances, any plasma protein that crosses the vascular wall cannot easily reach neurons in the brain. However, the increased cerebrovascular permeability, as we have shown here, enables Fg extravasation and deposition in the parenchyma, and this leads to a potential interaction with neurons. According to our previous findings, these specific interactions of Fg with neurons result in neuronal death [16,29]. The present data showed that there is an increased colocalization of Fg with neuronal axons and dendrites during TBI, which was lower in mice with a reduced content of Fg.

Since astrocytes are predominantly located between capillaries and neurons, it is obvious that extravasated proteins would be in contact with astrocytes first. Previously, we have shown that during TBI, after extravasation, Fg deposits in the vasculo-astrocyte interface and results in activation of astrocytes [10]. It has been shown that astrocytes phagocytose Fg [30]. However, in that study, it was shown that as a result of Fg and astrocyte interactions, the number of astrocytes decreases. This phenomenon was not discussed in [30]. Our studies show that a specific interaction of Fg with astrocytes lead to their activation [15,16] and their increased death [16]. Combined, these data indicate that extravascularly deposited Fg nevertheless results in the death of astrocytes, despite the fact that some of it can be phagocytosed by astrocytes, and as the inflammation persists during TBI, continuing extravasation of Fg facilitates the contact of Fg with neurons. Others have also shown the colocalization of Fg with neurons in a porcine model of TBI [12], a model that is more similar to human TBI. In the present study, we observed lower extravascular colocalization of Fg with neurons in mice with chronic hypofibrinogenemia compared to that in WT mice. This finding suggests that there is less extravasation of Fg in the Fg γ+/− mice.

Previously, we found that HFg was involved in the overexpression of the pro-inflammatory activating transcription factor 3 (ATF-3) gene [18]. Later, we tested the direct effect of Fg on cultured primary neuronal cells and found that Fg dose-dependently increased the expression of NF-кBp65 protein in neurons [17]. ATF-3 is induced by inflammatory and oxidative stress conditions, while NF-кB plays a significant role in inflammation, immunity, and regulation of apoptosis [31]. Direct binding and the association of ATF3 with NF-кBp65 in inflammatory conditions have been shown [31]. NF-кB is expressed in neurons and glia, and one of its diverse functions is to regulate the inflammatory reaction [32]. We have shown in vitro that the interaction of Fg with neurons not only induced the expression of NF-кBp65 [17] but also increased the expression of pro-inflammatory cytokine interleukin 6 and resulted in neuronal death [29]. However, Fg-induced expression of neuronal NF-кBp65 in the brain tissue has never been shown. The presented results indicate that there was lower colocalization of NF-кBp65 and neurons in the brain of Fg γ+/− mice compared to that in WT mice during TBI, suggesting that there was less activation of NF-кB in the neurons as a result of minimal interaction of Fg with neurons. These results suggest a strong role of Fg in the neuronal expression of the proinflammatory transcription factor and thus in neurodegeneration seen during m-mTBI [10,18]. Thus, reduced availability of Fg during inflammatory disease, such as TBI, may result in a reduction in neuroinflammation. We speculate that similar effects can be achieved by inhibiting interactions of Fg with its receptors on neurons. We have previously shown that blocking Fg interactions with its receptors, intercellular adhesion molecule-1 (ICAM-1) and PrP^C^, on neurons or astrocytes resulted in a reduction in the expression of astrocytic and neuronal pro-inflammatory genes and proteins and a reduction in Fg-induced oxidative damages, apoptosis, and cell death [16,17].

HFg has been shown to be associated with an increased risk of Alzheimer’s type dementia [33]. Extravascular deposits of Aβ, the hallmark pathology of Alzheimer’s disease, are found in the brains of patients after head trauma [34]. It is known that Aβ in brain tissue is often associated with Fg/fibrin deposits during AD [35]. Fibrin clots formed in the presence of Aβ are structurally abnormal and highly resistant to degradation [36]. It was shown that inhibition of an Aβ and Fg interaction resulted in improved cognitive deficit in AD transgenic mice [35]. We previously showed that the interaction of Fg with neurons occurs through its neuronal receptor PrP^C^ [29]. PrP^C^ is known to play a significant role in the cognitive function and memory deficit seen in TBI [37]. Thus, reducing the blood level of Fg during inflammatory diseases, such as TBI, may result in decreased formation of Fg-PrP^C^ complexes on the surface of neurons, and this can result in lower cognitive deficits seen during m-mTBI by reducing neurodegeneration. Combined, our studies indicate that reducing the level of Fg in the blood or blocking its interaction with its specific receptors on neurons can result in the mitigation of Fg-induced neuroinflammation and neurodegeneration and potentially reduce cognitive decline, as we have seen previously [16,18,29].

We have shown that Fg dose-dependently increased the activation of astrocytes [15,16]. Astrogliosis is a reaction of astrocytes due to an injury or pathological process in the CNS, with the hallmark characteristic of upregulation in GFAP [38]. In the current study, we found that the astrogliosis resulting from TBI was greatly reduced during hypofibrinogenemia. Since astrocytes constitute approximately 30% of the cells in the mammalian central nervous system [39], their activation could lead to deleterious effects, as it is known that reactive astrocytes are neurotoxic [40]. Astrocytes play a crucial role for normal functioning neurons [41,42]. They regulate synaptic transmission and modulate synaptic plasticity, long-term potentiation, and memory formation [43]. In the current study, we are the first to show that a decrease in blood levels of Fg caused a significant reduction in GFAP expression (one of the markers of astrocyte activation) [38]. These results suggest that lowered circulating Fg during TBI may result in a decrease in neurodegeneration and in the STM changes seen in previous studies [10,14,18,22].

Previously, we showed that inhibition of Fg synthesis with Fg antisense oligonucleotide (Fg-ASO) ameliorated enhanced cerebrovascular permeability and improved STM reduction in mice with CCI [18]. In that study, Fg-ASO was delivered post-injury in the form of several injections of necessary doses for the duration of the experiment in order to reduce the synthesis of Fg [18]. Although the use of Fg-ASO resulted in a decrease in Fg levels of the blood with low fluctuations [18], chronic reduction in the blood content of Fg in Fg γ+/− mice warranted a more steady and consistent low level of Fg. In the current study, we found that the reduction in Fg levels in the blood ameliorated the increased pial venular permeability in mice with CCI. The lowered cerebrovascular permeability is apparent both during the chronic reduction in the Fg level (Fg γ+/− mice) and during an acute reduction in Fg blood levels with Fg-ASO [18]. Combined, our data show that a reduction in the blood level of Fg can be beneficial for the reduction in cerebrovascular permeability and the resultant deposition of Fg in the brain tissue during TBI.

The Fg γ+/− mice that we used for the current study were Fg gamma-chain knockout mice [19,20], which are different from the Fg A-alpha-chain knockout mice (Fg-Aα-KO) developed by Dr. Jay L. Degen [44,45]. With the targeted deletion of the Aα chain, all of the component chains of Fg (Aα, Bβ and γ) are immunologically indetectable in the circulation, and the blood samples fail to clot or support platelet aggregation [45]. Similarly, the homozygous strain of Fg γ-chain knockout mice (Fg γ−/−) also shows indetectable circulating Fg with spontaneous neonatal bleeding. This is because the presence of all three Aα, Bβ and γ chains is a requirement of the Fg protein assembly as intermediates for the protein synthesis [5,46]. However, the heterozygous Fg γ+/− mice used in our experiments had a lower blood level of Fg, with no delay in blood clotting [20]. It is known that the Fg-γ chain sequence 117–133 interacts with ICAM-1 [47], which is a known Fg ligand [48,49]. We have shown that Fg interacts with ICAM-1 on astrocytes and neurons [16,29]. Therefore, the specific γ chain deletion in our Fg γ+/− mice permits the phenotypic analysis of our results with the assumption that there is a reduced Fg interaction overall, but an interaction of Fg with its receptor ICAM-1 is completely absent. Therefore, a possible interaction of Fg with neurons seen in the present study may occur through its other receptor, PrP^C^ [16].

Hypofibrinogenemia has been associated with some cardiovascular diseases [50,51]. Fibrates such as gemfibrozil and denofibrate are commonly used in patients with cardiac problems and severe hypertriglyceridemia [50]. Similarly, the use of ticlopide, a blood thinner in patients with peripheral arterial disease and cerebrovascular disease, was associated with the hemorheological modification of decreased plasma viscosities and resulted in clinical improvements [51]. We suggest that the benefit of lowered blood levels of Fg could be beneficial not only in the cases of cardiovascular disease, but also in Fg-induced neurodegeneration during TBI.

In conclusion, the present study demonstrated that lowering the blood level of Fg ameliorated the enhanced cerebrovascular permeability, GFAP expression, and neuronal pro-inflammatory NF-кBp65 expression during m-mTBI in mice. These results suggest the role of Fg in VCI and particularly in neurodegeneration, leading to the reduction in STM during TBI or other neuroinflammatory diseases associated with HFg.

## 4. Materials and Methods

### 4.1. Animals

The use of animals and the maintenance of their welfare adhered to the National Institutes of Health (NIH)’s Guide for the Care and Use of Laboratory Animals. Described here, protocols were reviewed and approved by the University of South Florida Animal Care and Use Committee. Twelve-week-old, 26–30 g male and female C57BL/6J WT and Fg γ+/− mice were used. The Fg γ+/− mice have a normal thrombin clot time similar to that observed in the WT mice [20].

Genotyping samples were collected using an ear-punch. Genotyping of Fg γ+/− mice was conducted by polymerase chain reaction analysis of purified genomic DNA from the collected tissue according to the described protocol [20]. In Fg γ+/− mice, the complete coding sequence of the Fg γ-chain was replaced with a *NEO* gene for identification. Therefore, the reaction mixture contained three primers: a *NEO* gene-specific primer (5′-GGTTCTAAGTACTGTGGTTTC-3′), a sequence within the 5′- of the non-coding flank of the Fg gamma chain (Fg-γ) (5′-CACAGCGGCTTGTCATTAG-3′), and a sequence internal to the Fg-γ (5′-CTGAAAGACCTGTCTTTGC-3′). The PCR products were 420 bp for the WT allele and 550 bp for the Fg-γ disrupted allele (Figure 1).

Mild-to-moderate TBI was generated according to the method described earlier [11] and routinely performed in our lab [9,14]. Briefly, mice were anesthetized with isofluorane and placed in a stereotaxic frame (Kopf, Tujunga, CA, USA). A 4 mm diameter craniotomy window was created with a high-speed microdrill (Harvard Apparatus, Holliston, MA, USA) at −2.5 mm bregma and 2.75 mm lateral to the midline over the left hemisphere. The bone disk was lifted using a microrongeur (Fine Science Tools, Foster City, CA, USA) to form a cranial window. The dura mater was left intact. The cranium was continuously irrigated with an artificial cerebrospinal fluid (Harvard Apparatus, Holliston, MA, USA) at 37 °C. TBI was induced using a CCI device. The impactor device (TBI 0310, Precision Systems & Instrumentation, Fairfax Station, VA, USA) with a 2 mm diameter flat tip was set to deliver an impact (0.5 mm impact depth, 3.5 m/s velocity, 500 ms) to the cortical surface. After the impact, the cranial window was covered using a Surgicel^®^ absorbable hemostat (Johnson & Johnson, Somerville, NJ, USA), the cranial bone was reimplanted, and the skin was sutured. Sham-operated animals were subjected to the same surgical procedure without delivery of the impact.

### 4.2. Permeability Study

Observation of microvascular permeability was performed 14 days after CCI or sham injury according to the method described previously [9,14]. Mice were anesthetized with ketamine (87.5 mg/kg)–xylazine (12.5 mg/kg, ip) for general anesthesia. A carotid artery was cannulated with polyethylene-10 tubing (Stoelting, Wood Dale, IL, USA) to allow for the injection of the FITC-BSA for observation of cerebrovascular permeability changes. The skull cap was removed, and the cranial window was extended using a micro-drill. The surface of the exposed pial circulation was kept moist by means of constant superfusion with phosphate-buffered saline (PBS). Following surgical preparation and a 30 min equilibration period, the mice were placed on a stereotaxic frame on the stage of an Olympus BXG61WI microscope (Olympus, Tokyo, Japan). Prior to the experiment, autofluorescence of the observed area was recorded. The FITC-BSA (stock solution 300 µg/mL) was infused (0.2 mL/100 g bd.wt.) through carotid artery cannulation connected to a syringe pump (Harvard Apparatus, Holliston, MA, USA) at 40 µL/min speed and allowed to circulate for 5 min. The pial circulation was surveyed to ensure the absence of spontaneous leakages. Cerebrovascular permeability was studied in the region of the injury penumbra, at least 200 µm away from the injury perimeter. Venules were identified by observing the topology of the pial circulation and blood flow direction. Images of selected third-order venular segments were recorded at baseline and in 15 min interval for an hour. The area of interest was exposed to blue light (488 nm) for 10–15 s. The microscope images were acquired by an electron-multiplying charge-coupled device camera (Quantum 512SC, Photometrics, Tucson, AZ, USA) and image acquisition system (Olympus CellSens Dimension Desktop 2.3). At the end of the experiment, mice were euthanized and their brains were immediately excised and flash frozen [52]. The frozen brains were cryosectioned and processed for immunofluorescence staining.

### 4.3. Western Blot

At the end of each experiment, blood was collected from the vena cava with a needle and syringe that was rinsed with sodium citrate anti-coagulant (10.9 mml/L) with 1 part anticoagulant to 9 parts blood. Hemolyzed blood samples were discarded. Blood samples were centrifuged at 1000× *g* for 5 min to obtain plasma. The plasma samples were kept frozen at −70 °C until analysis. The Bradford assay (Bio-Rad Laboratories, Inc., Hercules, CA, USA) was used for protein quantification following the protocol recommended by the manufacturer. Plasma samples were mixed with appropriate content of Laemmli buffer containing 2-mercaptoethanol (Bio-rad) and boiled for 3 min.

An equal amount of proteins were separated on a 7.5% Mini-Protean TGX (Tris-glycine extended) stain-free precast gel (Bio-Rad, cat #4568023) and transferred to nitrocellulose membranes. The membranes were immunoblotted using the rabbit anti-Fg antibody at 1:40,000 (Agilent Dako, Santa Clara, CA, USA, cat. #A008002-2) or mouse anti-β actin antibody at 1:20,000 (Millipore sigma, Burlington, MA, USA, cat. #A3854-200UL). The secondary antibodies used were goat anti-rabbit IgG (H+L), horseradish peroxidase (HRP) (Invitrogen cat. #A16110) and goat anti-mouse IgG (H+L), HRP (Invitrogen cat. #A16072), at 1:40,000. The proteins were detected with Pierce ECL Western Blotting Substrate (Thermo Fisher, Waltham, MA, USA, cat. #32109). Immunoblots were developed using a ChemiDoc XRS+ imager system (Bio-Rad) and quantified by means of densitometric analysis using Image Lab 6.0.1. The results were presented as a ratio of the IOD of Fg bands to the IOD of the respective β-actin bands.

### 4.4. Immunofluorescence Staining

A separate group of animals that were not used for the cerebrovascular permeability study were sacrificed fourteen days after CCI or sham surgery. Ketamine–xylazine-anesthetized animals were slowly infused with 100 µL of Texas Red dye-labeled lycopersicon esculentum lectin (LEA, Vector Laboratories Inc., Burlingame, CA, USA) through the jugular vein. After a 20 min equilibrium period, the mice were infused with 4% paraformaldehyde (PFA) in PBS solution through the left ventricle. The mouse brain tissue was processed as previously described [10,14,22]. The brains were collected and placed in 4% PFA overnight, followed by incubation in 30% sucrose for 3 days. Brain samples were embedded in a protective matrix (Thermo Scientific Richard-Allan Scientific Neg-50, cat. # 6506), cryosectioned at 25 µm thickness with a Cryostar NX70 (Thermo Fisher Scientific, Waltham, MA, USA) and mounted on a gelatin-coated self-adhesive microscope slide (Thermo Fisher Scientific).

The brain samples were blocked with a blocking buffer (1X PBS/5% goat serum/0.3% Triton^TM^ X-100) for 1 h at room temperature (RT). After washing with PBS, the ipsilateral hippocampus of the experimental animals was incubated with primary antibodies overnight at 4 °C. The primary antibodies used were rabbit anti-Fg at 1:200 (Dako, cat. # A008002-2), mouse anti-GFAP at 1:500 (Proteintech, Rosemont, IL, USA, cat. #60190-1-1g) to label astrocytes, rabbit anti-NF-кBp65 at 1:150 (Millipore sigma, cat. #06-418), and to label neurons, chicken anti-microtubule-associated protein 2 at 1:5000 (Thermo Fisher Scientific, cat. #PA1-10005) and neuronal nuclear (NeuN) antibody at 1:50 (Millipore sigma, cat. #MAB377) were used. After washing with PBS three times, for 5 min each, tissues were incubated with respective Alexa Fluor-conjugated secondary antibodies (Thermo Fisher Scientific) at 1:200 for 2 h at RT.

### 4.5. Confocal Microscopy

The samples stained with immunofluorescence were analyzed using an Olympus FV 1200 laser-scanning confocal microscope (Olympus, Tokyo, Japan). Four areas of interest (AOI) per sample were examined at least 200 µm from the perimeter of the impact for Figure 3 and Figure 4, while in Figure 5, images from the hippocampus of the ipsilateral portion are presented. The microscope settings were kept constant. The acquired images were analyzed with Olympus CellSens Dimension Desktop 2.3 (Olympus). The obtained data were averaged for each experimental group.

### 4.6. Data Analysis

All data were expressed as mean ± SEM. The experimental groups were compared using Student’s t-test (Figure 1b,d) or one-way ANOVA (Figure 2B, Figure 3B,C, Figure 5B and Appendix A). If an ANOVA indicated a significant difference (*p* < 0.05), Tukey’s multiple comparison test was used to compare group means. The differences were considered significant if *p* < 0.05. The degree of colocalization of MAP2 and Fg in Figure 4 was assessed by measuring *R* and Mander’s coefficients and presented in Table 1.

## Figures and Tables

**Figure 1 biomolecules-14-00385-f001:**
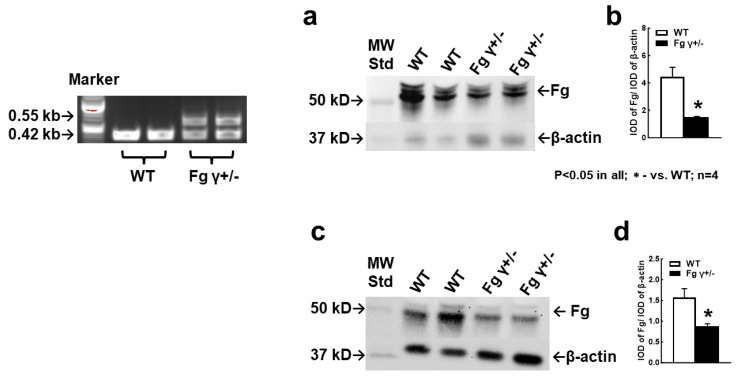
Genotyping of wild-type (WT) and heterozygous Fg gamma-chain-knock out (Fg γ+/−) mice. In Fg γ+/− mice, the complete coding sequence of the Fg γ-chain was replaced with a *NEO* gene for identification. Amplification of the disrupted allele leads to a 0.55 kb amplicon, whereas amplification of the wild-type Fgγ-allele gene results in a 0.42 kb amplicon. Dual polymerase chain reaction products suggest the heterozygous mutation of Fg gene (Fg γ+/−), while their absence represents wild-type (C57BL6) alleles. **Inset**: Western blot (WB) analyses for content of Fg in the plasma from WT and Fg γ+/− mice 14 days after the sham operation (Sham, **a**,**b**) and cortical contusion injury (CCI, **c**,**d**). (**a**,**c**) Examples of WB images; (**b**,**d**) summaries of respective data analyses. *p* < 0.05 in all; *—vs. WT; *n* = 4.

**Figure 2 biomolecules-14-00385-f002:**
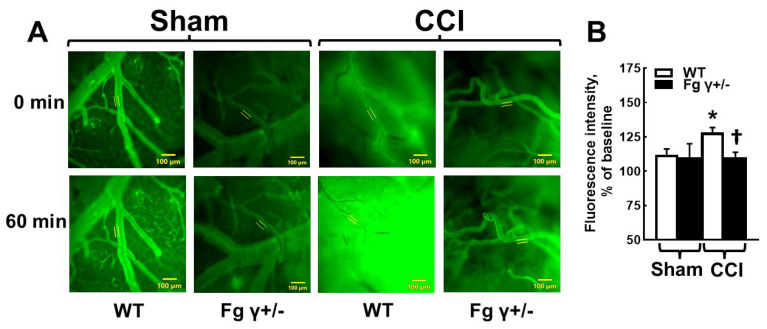
Macromolecular leakage of mouse pial venules 14 days after cortical contusion injury (CCI). (**A**) Representative images of microvessels with infused fluorescein isothiocyanate-conjugated bovine serum albumin (FITC-BSA) in the wild-type (WT) and heterozygous Fg gamma-chain-knock out (Fg γ+/−) mice after the sham operation or CCI) and after 1 h of the post equilibrium period. Two lines profile probes (LPP) (appear as yellow lines) were placed inside and outside of the area of interest of the vessel to measure the fluorescence intensity. (**B**) The average ratios of fluorescence intensities outside the vessel to those inside of the vessel are presented as a percentage of the baseline. *p* < 0.05 for all; *—vs. WT-Sham, †—vs. WT-CCI; *n* = 6.

**Figure 3 biomolecules-14-00385-f003:**
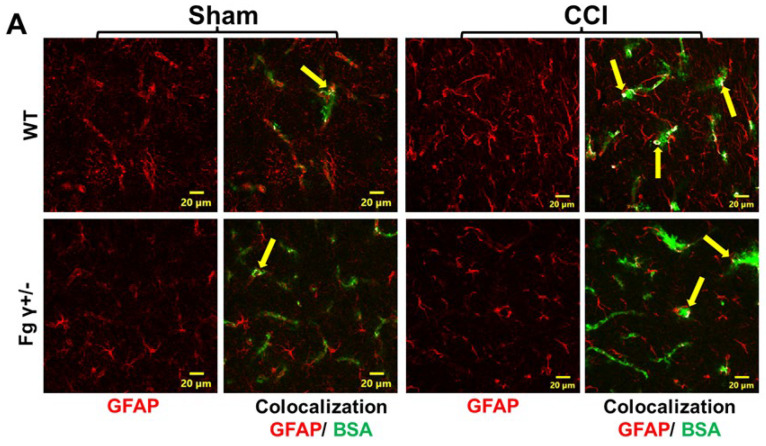
Extravascular deposition of bovine serum albumin (BSA) and the expression of glial fibrillary acidic protein (GFAP) in the ipsilateral cortical brain sections 14 days after the sham operation (Sham) or cortical contusion injury (CCI). (**A**) Representative images show the extravasated fluorescein isothiocyanate-conjugated BSA (FITC-BSA, green) co-localized with activated astrocytes marked with GFAP (red) in the wild-type (WT) and heterozygous Fg gamma-chain-knock out (Fg γ+/−) mice. The yellow arrow identifies the colocalization spots of extravasated FITC-BSA and astrocytes. (**B**) Summary of GFAP fluorescence intensity in the WT and Fg γ+/− after CCI and the sham operation. (**C**) Summary of FITC-BSA and GFAP colocalization spot counts. *p* < 0.05 for all; *—vs. WT-Sham, †—vs. WT-CCI; *n* = 4.

**Figure 4 biomolecules-14-00385-f004:**
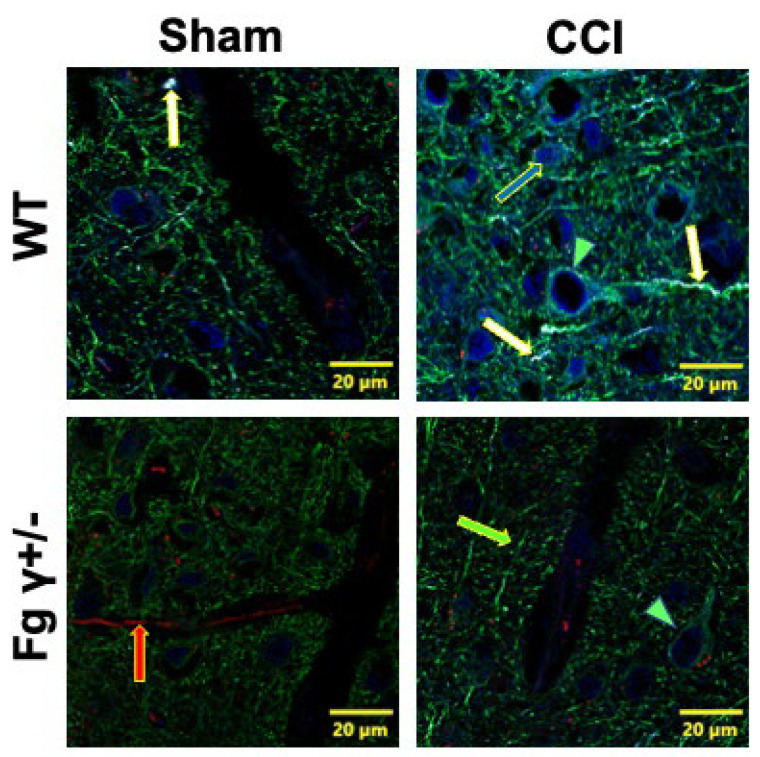
Deposition of fibrinogen (Fg) and its colocalization with neurons in the wild-type (WT) and Fg gamma-chain-knock out (Fg γ+/−) mice 14 days after the sham operation or cortical contusion injury (CCI). Representative images of colocalization of Fg (blue) with neurons defined with microtubule-associated protein 2 (MAP2, green). The red arrow shows lycopersicon esculentum agglutinin-labeled blood vessels; the green arrow shows MAP2-labeled neural axons and dendrites; the green arrowheads point to the neural soma; the blue arrow indicates Fg; and the white arrows identify colocalization of Fg and neurons.

**Figure 5 biomolecules-14-00385-f005:**
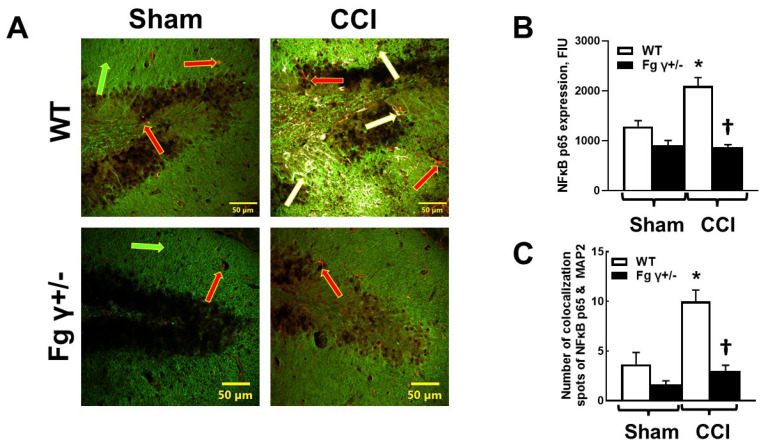
Expression of nuclear factor-кB (NF-кB) p65 in the wild-type (WT) and heterozygous Fg gamma-chain-knock out (Fg γ+/−) mice after the sham operation (Sham) or cortical contusion injury (CCI). (**A**) Representative images of NF-kBp65 expression and its colocalization with neurons defined with microtubule-associated protein 2 (MAP2). Red arrows indicate NF-kBp65 in neurons, green arrows indicate microtubule-associated protein 2 (MAP2)-labeled neural dendrites, and white arrows indicate the colocalization of NF-kBp65 and MAP2. (**B**) Summary of fluorescence intensity changes of NF-кBp65 in WT and Fg γ+/− mice after CCI or sham operation. (**C**) Summary of NF-kBp65 and MAP2 colocalization spot counts *p* < 0.05 for all; *—vs. WT-Sham, †—vs. WT-CCI; *n* = 4.

**Table 1 biomolecules-14-00385-t001:** Analysis of colocalization of fibrinogen (Fg) with neurons in the cortex of wild-type (WT) and Fg gamma-chain knockout (Fg γ+/−) mice after cortical contusion injury (CCI) or a sham operation.

Treatment Groups	WT-Sham	WT-CCI	Fg γ+/−-Sham	Fg γ+/−-CCI
Total area, μm^2^	9.6 ± 3.7	63.7 ± 2.5	0.2 ± 0.2	23.0 ± 4.8
Area Fraction, %	0.013 ± 0.008	0.11 0 ± 0.004	0.033 ± 0.020	0.030 ± 0.012
R(r)	0.20 ± 0.05	0.44 ± 0.05	0.135 ± 0.141	0.103 ± 0.072
M1	0	0.008 ± 0.003	0.003 ± 0.003	0
M1	0.018 ± 0.005	0.105 ± 0.023	0.030 ± 0.015	0.043 ± 0.012

R(r)—Pearson’s correlation coefficient; M (1)—colocalization coefficient describes the contribution of the channel 1 fluorophore to the colocalized area; M (2)—colocalization coefficient describes the contribution of the channel 2 fluorophore.

## Data Availability

Data are contained within the article and Appendix A.

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
