# Peer review of "The Effect of Reduced Fibrinogen on Cerebrovascular Permeability during Traumatic Brain Injury in Fibrinogen Gene Heterozygous Knockout Mice"

_biomolecules, 2024, doi:10.3390/biom14040385_

Round 1

Reviewer 1 Report

Comments and Suggestions for Authors
The authors have built upon previous work to understand the effects of Fg-induced neuroinflammation following blood-brain permeability during mild to moderate CCI in a transgenic mouse model. The rationale for the study is well-established and relevant to current literature on the topic.

The authors conducted western blots to measure Fg expression in the blood following sub-acute TBI and sham surgery. Brain tissue IHC was performed to analyze GFAP expression, Fg deposition with MAP2, and nuclear factor KBp65.

 Comments: 
1- Discrepancies between Western blot (WB) and immunohistochemistry (IHC) results are confusing. While WB reveals reduced Fg expression in TG mice compared to WT following both CCI and sham surgeries, IHC indicates comparable findings between TG sham, WT sham and TG TBI groups. The authors are encouraged to elucidate the reasons behind this disparity. Given the anticipated lower Fg levels in TG sham compared to WT, it raises questions about the observed similarity in pathology between TG sham, WT sham, and Fg TBI. Please clarify the western blot results accordingly.

2- The authors need to present brain slices images of the injury site to confirm that the craniotomy has not caused any damage to the WT group, hence increasing Fg compared to sham shown in the western blot analysis.

3- Behavioral tests such as water maze for cognition, rotarod and open field tests for motor deficits should be performed to confirm CCI injury and measure the functional aspect of positive changes, especially considering the link between Fg-induced neuroinflammation and cognitive decline mentioned in the discussion.

4- The authors should specify the number of animals used per group to validate any significance.

5- Explanation is needed regarding the criteria for identifying activation of astrocytes. If changes in morphology were measured, 25μm slices may not cover the entire GFAP, thereby potentially missing dendritic changes in astrocytes. The authors should consider changing "activation" to "increase in GFAP expression" in the manuscript and either in method or supplementary explain the method of GFAP analysis.

6- The authors should justify why a 14-day sub-acute recovery was chosen as a single timepoint. How does Fg levels following a 2 week recovery compare to acute and chronic stages.

7- Clarification is needed regarding the difference in p-value notation between * and T in the figures.

8- Supplementary figures and original images are the same document. The original western blot image of Figure 1B should be changed to 1A.

Reviewer 2 Report

Comments and Suggestions for Authors

In this manuscript, the authors investigated the effect of a chronic reduction of fibrinogen level in blood on cerebrovascular permeability and the interactions of extravasated fibrinogen with astrocytes and neurons. They performed cortical contusion injury (CCI) to generate mild-to moderate traumatic brain injury (m-mTBI) in wild type (WT) and transgenic heterozygous mice with a deleted fibrinogen γ chain resulting in a low blood level of fibrinogen. They found that increased fibrinogen extravasation, and its resultant deposition parenchymal tissues, triggered astrocyte activation and lead to potential interactions of fibrinogen with neurons resulting in the overexpression of neuronal NF-кB following 14 days m-mTBI, and suggested that reduced level of blood fibrinogen could be beneficial in mitigating short-term memory (STM) reduction in m-mTBI. The manuscript is well-written, well-designed, and interesting; however, the authors need the following revision to improve the quality of this study:

1)     Exchange the quantitative data and images for sham groups first followed by CCI groups in all corresponding figures.

2)     Figure 3: Mention the staining color for GFAP (red) and FITC-BSA (green) in the figure legend.

3)     Figure 4: Provide double staining with neuronal nuclei (NeuN) to confirm whether fibrinogen is colocalized with neuronal cell bodies or its exons.

4)     Figure 5: Provide separate and merge images including high magnification images of MAP2 and NF-kBp65 to clearly show the colocalization of NF-kBp65 with neurons. It looks poor MAP2 staining in Fg γ+/- mice after CCI compared to other groups; please replace or explain it. The significant sign “† - vs. WT TBI” is mentioned in figure legend but absent in corresponding quantitative data.
